# Factors Associated with Unplanned Transfer of Patients with Brain Tumor from Inpatient Rehabilitation Unit to Primary Acute Care Units

**DOI:** 10.3390/jpm13010131

**Published:** 2023-01-10

**Authors:** Gyoung Ho Nam, Won Hyuk Chang

**Affiliations:** 1Department of Physical and Rehabilitation Medicine, Center for Prevention and Rehabilitation, Heart Vascular and Stroke Institute, Samsung Medical Center, Sungkyunkwan University School of Medicine, Seoul 06351, Republic of Korea; 2Department of Health Sciences and Technology, Department of Medical Device Management & Research, Department of Digital Health, SAIHST, Sungkyunkwan University, Seoul 06355, Republic of Korea

**Keywords:** rehabilitation, brain tumor, complication, patient transfer

## Abstract

Inpatient rehabilitation should be assessed to improve each functional domain in patients with brain tumor. However, no previous study has reported risk factors for unplanned transfer of this patient population to primary acute care units during a comprehensive inpatient rehabilitation. The objective of this study was to investigate the percentage of unplanned transfer of brain tumor rehabilitation inpatients to primary acute care units compared with stroke patients and factors associated with such unplanned transfer. Data of 137 patients with brain tumor who were transferred to the department of physical and rehabilitation medicine were retrospectively reviewed. For comparison, data of 438 patients with subacute stroke were also obtained. Included patients were divided into an unplanned transfer group and a control group based on whether they required a transfer to another department for acute care before completing their comprehensive inpatient rehabilitation. Reasons for unplanned transfers were classified based on medical or surgical conditions. The incidence of unplanned transfers to the medical or surgical department was significantly higher in patients with brain tumor (15.3%) than in stroke patients (7.1%) (*p* < 0.05). Most of unplanned transfers occurred within two weeks of the comprehensive inpatient rehabilitation for patients with brain tumor. There was a significantly higher incidence of unplanned transfers in patients with a primary tumor than in those with a metastatic tumor (15.9% vs. 4.8%, *p* < 0.05). In addition, the frequency of chemotherapy or radiotherapy was significantly (*p* < 0.05) higher in the unplanned transfer group than in the control group. The most common cause of an unplanned transfer was a neurologic cause (90.0%) in patients with brain tumor and an infectious disease such as pneumonia (51.6%) in stroke patients. In conclusion, these results demonstrated a higher incidence of unplanned transfers in patients with brain tumor than in stroke patients during intensive inpatient rehabilitation. Proportions of those with neurological problems were relatively higher in patients with brain tumor than in patients with subacute stroke.

## 1. Introduction

Recent advances in chemotherapy, radiation therapy, and surgery have significantly lengthened life expectancy of patients with brain tumors [1]. Despite these advances in treatment, functional impairments due to neurologic sequelae can occur in many brain tumor survivors [1]. Brain tumor itself causes problems such as tumor progression or metastasis; therefore, various neurologic sequelae can occur in the process of treating the tumor, including surgical complications and neurotoxic effects of radiation and chemotherapy [2,3]. Therefore, some patients with brain tumor require an intensive inpatient rehabilitation service for functional improvement and quality of life [1,4]. 

However, these patients are more likely to develop unexpected medical problems that can lead to returning of patients to acute care units or transfer of them to other departments during a comprehensive inpatient rehabilitation. Such return or transfer often prevents them from receiving timely rehabilitation treatment, thus increasing the length of hospitalization, mortality, and economic burden of patients and their family [5]. Therefore, a better understanding of factors associated with transfer to acute care units is needed to help identify high-risk patients and increase the potential of more effective care and cost savings. Although incidences of unexpected medical problems and risk factors for returning to primary acute care units during a comprehensive inpatient rehabilitation in subacute stroke patients have been reported [6], the incidence and risk factors for returning to primary acute care units in patients with brain tumor during a comprehensive inpatient rehabilitation have not been reported yet. 

Thus, the purpose of this study was to investigate the incidence of patients with brain tumor with an unplanned transfer to the primary acute care department during a comprehensive inpatient rehabilitation and factors associated with such unplanned transfer.

## 2. Materials and Methods

### 2.1. Participants

Retrospective medical data of patients with brain tumor transferred to the department of physical and rehabilitation medicine (PRM) for comprehensive intensive inpatient rehabilitation in a tertiary university hospital from August 2018 to August 2021 were obtained. For comparison, data of patients with early subacute stroke (7 days to 3 months from the date of onset) [7] were also obtained. The brain tumor group included patients with all types of primary solid brain tumor, hematologic cancer, and secondary metastatic tumors. The stroke group included patients with all types of cerebral infarction and hemorrhage. The severity of stroke and brain tumor was not considered for each group. Patients aged less than 19 years were excluded from analysis.

All included patients with brain tumor and subacute stroke were divided into an unplanned transfer group and a control group based on whether they required a transfer to another department for acute care before completing their comprehensive intensive inpatient rehabilitation. Reasons for unplanned transfers were classified based on unexpected worsening of serious neurological and/or medical conditions. Neurological conditions included central nervous system (CNS) infection, recurrence of stroke, brain tumor aggravation, brain edema, and hydrocephalus. Medical conditions included infections except CNS (e.g., pneumonia, colitis, etc.), cardiopulmonary diseases, gastrointestinal diseases, and renal problems. Regarding medical problems, the number of patients was too small to subdivide causes of infection. Thus, other infectious diseases except pneumonia were grouped into one group for convenience of classification. The control group included patients who were discharged from the hospital and patients with a planned transfer for scheduled further management after successfully completing rehabilitation treatment.

### 2.2. Demographic Data

Groups were compared in terms of sex, age, functional status, length of stay from admission or transfer to the department of PRM to discharge, and past medical history. Initially, the Korean-National Institute of health stroke scale (K-NIHSS) [8] and Glasgow Coma Scale (GCS) [9] were checked for the stroke group. In the brain tumor group, history of surgical management, chemotherapy, and radiation therapy were investigated. In order to evaluate the functional status at baseline, the Korean version of the modified Barthel Index (K-MBI) [10], Mini-mental status examination (K-MMSE) [11], Functional ambulation category (FAC) [12], and Motricity index (MI) [13] were performed or determined at the time of transfer or admission to the Department of PRM and at the time of discharge or transfer to another department [1].

### 2.3. Statistical Analysis

Data were imported into SPSS version 27.0 (IBM, Armonk, NY, USA) for analysis. The chi-square test was used for comparing categorical variables. The Shapiro–Wilk test was used to determine the distributional normality of all continuous variables (all were found to be normally distributed; *p* > 0.05). The independent t-test was used for comparing continuous variables. The unplanned transfers rates were compared between the brain tumor group and the stroke group using the Kaplan–Meier regression analyses. Statistical significance was defined at *p*-value < 0.05.

## 3. Results

A total of 137 patients with brain tumor and 438 patients with subacute stroke were identified based on the inclusion and exclusion criteria. General characteristics of each patient group (brain tumor and stroke) are presented in Table 1. The brain tumor group showed a significantly younger age than the stroke group (*p* < 0.05). The rate of comorbidity was significantly higher in the stroke group than the brain tumor group (*p* < 0.05). However, there was no significant difference in the functional status at admission to the Department of PRM between the two groups.

The incidence of unplanned transfers was significantly higher in the brain tumor group (*n* = 20, 14.6%) than in the stroke group (*n* = 31, 7.1%) (*p* = 0.010, Figure 1). In addition, the Kaplan–Meier regression analyses significantly showed a difference between the two groups in the probability of unplanned transfers during the comprehensive intensive inpatient rehabilitation (*p* = 0.007, Figure 2). Reasons for the unplanned transfer are listed in Table 2. The most common cause of an unplanned transfer was neurologic cause (90.0%) in the brain tumor group. However, medical causes such as pneumonia (51.6%) were the most common causes in stroke patients. Most unplanned transfers occurred within two weeks of the comprehensive intensive inpatient rehabilitation in the brain tumor group. On the other hand, the incidence of unplanned transfers according to duration of the comprehensive inpatient rehabilitation in the stroke group tended to show no difference (Figure 2).

In patients with brain tumor, there was no significant difference in age or sex between the unplanned transfer group and the control group. There was a significantly higher incidence of unplanned transfers in patients with a primary tumor than in those with a metastatic tumor (15.9% vs. 4.8%, *p* < 0.05). The incidence of chemotherapy was significantly higher in the unplanned transfer group than in the control group of patients with brain tumor (*p* < 0.05, Table 3). In stroke patients, there was a significantly higher incidence of supratentorial lesion in the unplanned transfer group than in the control group (*p* < 0.05). In addition, K-MBI, FAC, and K-MMSE were significantly lower in the unplanned transfer group than in the control group (all *p* < 0.05, Table 4).

## 4. Discussion

In this study, the incidence of an unplanned transfers during the comprehensive intensive inpatient rehabilitation was a significantly higher in patients with brain tumor than stroke patients (14.6% vs. 7.1%). Most unplanned transfers occurred within two weeks. The most common cause of an unplanned transfers was neurologic cause in patients with brain tumor. In addition, proportions of those with neurological problems were relatively higher in patients with brain tumor than in patients with subacute stroke. These results could be useful to identify patients with brain tumor at higher risk for unplanned transfer to the primary acute care units and to guide physiatrists for the closer supervision during the inpatient rehabilitation in patients with brain tumor.

Patients with tumor are more likely to require a transfer than patients without tumor, as shown in a previous report [14]. It has been shown that the increased risk of transfer is greater for patients with tumor with neurologic manifestations [14]. However, in this previous report [14], patients with various tumors, not only brain tumor, were recruited. Previous published studies have analyzed factors associated with return to the primary acute care inpatient service in patients with lymphoma [15], leukemia [16], hematopoietic stem cell [17], or multiple myeloma [18] during inpatient rehabilitation. The rate of transfer back to the primary acute care service in patients with multiple myeloma was lower than that in a previous hematologic malignancy population [18]. Bhakta et al. [19] recently reported the incidence of unplanned transfers in patients with brain tumor as 34%. They reported the incidence and factors of unplanned transfers according to the type of cancer. However, the comparison with subacute stroke, which is the target disorder of intensive inpatient neurorehabilitation, has not been reported to the best of our knowledge. In the present study, 14.6% of patients with brain tumors were transferred to acute care units, which was significantly higher than that in subacute stroke patients. In addition, neurologic problems were common causes for an unplanned transfer in patients with brain tumor. Among neurologic problems, hydrocephalus was the most common cause, followed by brain edema, tumor aggravation, and CNS infection. Factors associated with an increased risk of unplanned transfer to acute care units included primary brain tumor and history of chemotherapy. On the other hand, age and functional level of patients in the department of PRM showed no factor associated with an unplanned transfer. This higher incidence of unplanned transfers in patients with brain tumor could be due to their neurological fragility. Therefore, clinicians caring for rehabilitation inpatients with brain tumor should maintain close contact with acute neurological care units. 

Among patients with various brain disorders such as stroke, brain tumor, Parkinson’s disease, and traumatic brain injury, the effects of comprehensive inpatient rehabilitation on patients with subacute stroke have the clearest evidence. In addition, our previous study [1] reported that the effects of comprehensive inpatient rehabilitation in patients with subacute stroke and brain tumor were similar. Therefore, patients with subacute stroke were chosen and analyzed as a control group for patients with brain tumor in this study. Patients with brain tumor were younger than patients with subacute stroke in this study, similar to findings of previous studies [20,21] showing that the mean age was 53.3 years for those diagnosed with a brain tumor and 69.2 years for those diagnosed with a stroke. Proportions of comorbidities such as hypertension, diabetes mellitus (DM), and hyperlipidemia were higher in patients with subacute stroke, consistent with previous study results showing that stroke-related comorbidities were the most common in the order of hypertension, DM, and hyperlipidemia [22,23]. There was no significant difference in functional level between the two groups in this study. These results might also reflect the fact that the brain tumor itself has a higher neurological fragility than stroke. In previous studies [6,24], the most common cause of an unplanned transfer was medical problem and the second cause was a neurological problem in stroke patients. Common causes of unplanned transfers in stroke patients found in the present study were similar to those of previous studies [6,24]. The incidence of medical problems such as pneumonia was higher in the subacute stroke patient group than in the brain tumor group, which could be related to the fact that stroke patients were relatively older and accompanied by more comorbidities than patients with brain tumor [23].

As for the timing of transfer, subacute stroke patients showed a relatively uniform appearance at one month. However, brain tumor patients had a high rate of unplanned transfers within the first two weeks after admission to the department of PRM. It might be related to the characteristics of the department of PRM at a tertiary hospital in Korea [6]. In tertiary hospitals in Korea, most patients with brain tumor are transferred to the department of PRM within 1 to 2 weeks for early intensive comprehensive rehabilitation after primary treatment. Afterwards, additional treatments such as chemotherapy and radiation therapy are performed concurrently with intensive comprehensive rehabilitation. These could be more likely to make subacute complications occur within the early first 1–2 weeks [2,4]. Therefore, the multidisciplinary team should provide intensive comprehensive rehabilitation for patients with brain tumor. At the same time, it is necessary to anticipate and prepare for possible complications in consideration of their therapeutic history. Patients with brain tumor might need to be transferred to other rehabilitation institutes after stabilizing their medical condition for at least two weeks in the department of PRM at a tertiary hospital.

This study had some limitations. First, functional outcomes of patients with brain tumor could be influenced by many factors, such as tumor size, location, pathologic type, and previous specific cancer treatment history [25]. In this study, patients with brain tumor were classified only as those with primary, hematologic, and metastatic tumors. Therefore, we could not investigate each pathologic type or grade of tumor in detail. Brain lesions were classified into supratentorial, infratentorial, and both lesions. No exact brain lesion could be one of the limitations of this study. Future studies with additional characteristics will be needed. Second, the number of patients included in the study was relatively small. Although this study was conducted for 36 months, only 51 patients were included in the unplanned transfer group for both patients with brain tumor and subacute stroke. Although the most common cause of unplanned transfers in patients with brain tumor in this study was similar to that in a previous study [26], it was difficult to analyze infectious causes except pneumonia in detail due to a relatively small number of patients. Therefore, further studies with more patients will be need. 

## 5. Conclusions

Patients with brain tumor had a higher incidence of an unplanned transfers than stroke patients during intensive inpatient rehabilitation. The most common cause was a neurological problem in brain tumor patients. Since the incidence of unplanned transfers was high during the first two weeks, transfer to other rehabilitation institutes might be recommended after at least two weeks of medical stability in patients with brain tumor.

## Figures and Tables

**Figure 1 jpm-13-00131-f001:**
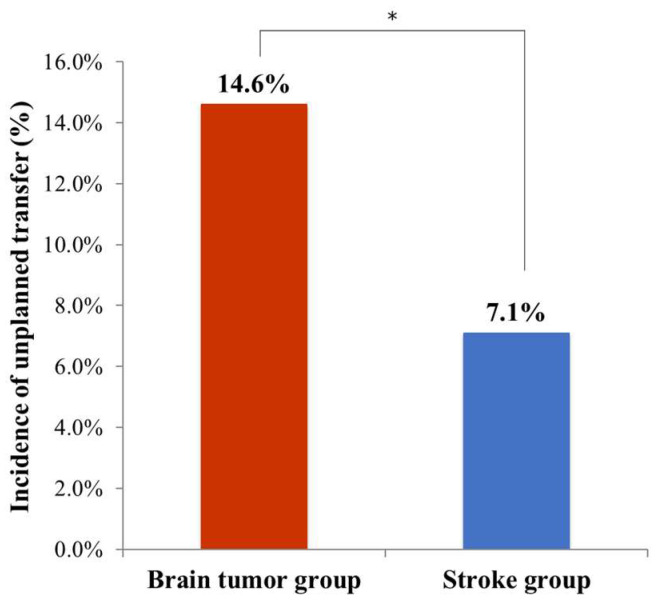
Incidence of unplanned transfers in each brain tumor and stroke groups. The incidence rate of unplanned transfers to acute care units was significantly higher in patients with brain tumor (*n* = 20, 14.6%) than in patients with stroke (*n* = 31, 7.1%) (*p* < 0.05). * *p* < 0.05, comparison between the two groups.

**Figure 2 jpm-13-00131-f002:**
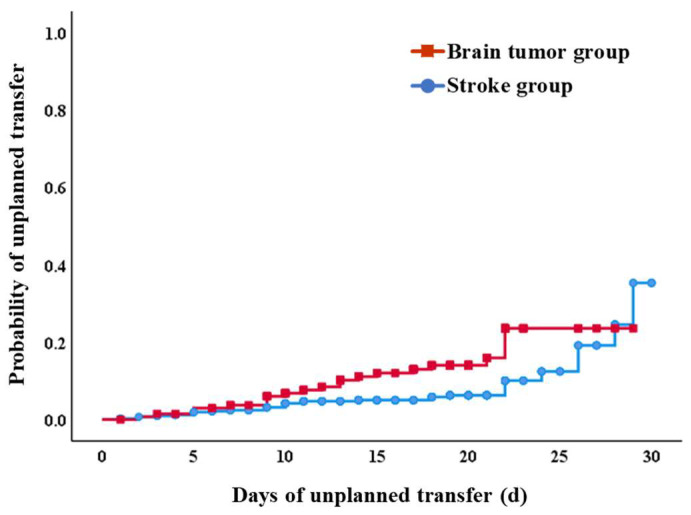
Probability of unplanned transfers in each group (brain tumor or stroke). The Kaplan–Meier regression analyses significantly showed on difference between the two groups in the probability of unplanned transfers during the comprehensive intensive inpatient rehabilitation (*p* < 0.05).

**Table 1 jpm-13-00131-t001:** General and functional characteristics of patients with brain tumor or stroke.

	Brain Tumor Group (*n* = 137)	Stroke Group (*n* = 438)	*p*-Value
Age, yr	56.9 ± 15.1 *	65.5 ± 15.3	<0.001
Sex (M:F)	77:60	234:204	0.624
Brain lesion (Supratentorial:Intratentorial:both)	86:35:16 *	194:195:49	<0.001
Lesion side (right:left:bilateral)	58:54:25	199:159:80	0.779
Comorbidity			
Hypertension (%)	29.9% *	57.9%	<0.001
Diabetes mellitus (%)	16.8% *	33.6%	<0.001
Atrial fibrillation (%)	4.4%	12.4%	0.006
Hyperlipidemia (%)	8.8% *	30.7%	<0.001
Functional status at admission to the department of physical and rehabilitation medicine			
K-MBI	28.2 ± 21.9	28.4 ± 26.3	0.927
MI-S	57.9 ± 21.5	59.1 ± 26.1	0.593
MI-U	60.9 ± 22.7	59.6 ± 27.7	0.608
MI-L	55.0 ± 24.5	58.5 ± 56.7	0.152
FAC	0.9 ± 1.1	1.1 ± 1.3	0.233
K-MMSE	18.2 ± 9.6	16.5 ± 10.7	0.069
Duration of inpatient rehabilitation, d	17.0 ± 5.7	17.9 ± 7.4	0.232

K-MBI, Korean version of the modified Barthel index; MI-S, side score of Motricity Index; MI-A, arm score of Motricity Index; MI-L, leg score of Motricity Index; FAC, Functional Ambulatory Category; K-MMSE, Korean Mini-Mental Status Examination. * *p* < 0.05, compared with stroke group.

**Table 2 jpm-13-00131-t002:** Reasons for unplanned transfers in each group.

	Brain Tumor Group (*n* = 20)	Stroke Group (*n* = 31)
Neurological causes (*n*, %)	18 (90.0%)	14 (45.2%)
CNS infection (*n*, %)	1 (5.6%)	0 (0.0%)
Recurrence of stroke (*n*, %)	-	8 (57.1%)
Tumor aggravation (*n*, %)	4 (22.2%)	-
Brain edema (*n*, %)	2 (11.1%)	0 (0.0%)
Hydrocephalus (*n*, %)	11 (61.1%)	6 (42.9%)
Medical causes (*n*, %)	2 (10.0%)	16 (51.6%)
Pneumonia	1 (50.0%)	6 (37.5%)
Infectious causes except pneumonia	0 (0.0%)	6 (37.5%)
Respiratory failure	0 (0.0%)	1 (6.3%)
Renal causes	0 (0.0%)	1 (6.3%)
Hemato-oncological	0 (0.0%)	1 (6.3%)
Colitis	1 (50.0%)	1 (6.3%)
Other causes (*n*, %)	0 (0.0%)	1 (3.2%)
Pressure ulcer	0 (0.0%)	1 (3.2%)

**Table 3 jpm-13-00131-t003:** Comparison between the unplanned transfer group and the control group in brain tumor patients.

	Unplanned Transfer (*n* = 20)	Control (*n* = 117)	*p*-Value
Age, yr	57.7 ± 14.0	56.8 ± 15.3	0.785
Sex (M:F)	14:6	63:54	0.226
Type of tumor *			0.028
Primary brain tumor (*n*, %)	18 (90.3%)	95 (81.2%)	
Metastatic brain tumor (*n*, %)	1 (5.0%)	20 (17.1%)	
Hematologic brain tumor (*n*, %)	1 (5.0%)	2 (1.7%)	
Brain tumor therapy			
Surgery (*n*, %)	17 (85.0%)	99 (84.6%)	1.000
Chemotherapy (*n*, %)	11 (55.0%) *	26 (22.4%)	0.005
Radiotherapy (*n*, %)	9 (45.0%)	27 (23.3%)	0.055
Functional status at admission to the department of physical and rehabilitation medicine			
K-MBI	23.6 ± 18.7	29.0 ± 22.4	0.253
MI-S	61.8 ± 23.3	57.3 ± 21.2	0.424
MI-U	66.3 ± 22.7	59.9 ± 22.6	0.257
MI-L	57.3 ± 27.0	54.6 ± 24.1	0.682
FAC	0.8 ± 1.1	1.0 ± 1.2	0.455
K-MMSE	18.2 ± 9.6	16.5 ± 10.7	0.069

K-MBI, Korean version of the modified Barthel index; MI-S, side score of Motricity Index; MI-A, arm score of Motricity Index; MI-L, leg score of Motricity Index; FAC, Functional Ambulatory Category; K-MMSE, Korean Mini-Mental Status Examination. * *p* < 0.05, compared with control group.

**Table 4 jpm-13-00131-t004:** Comparison between the unplanned transfer group and the control group in stroke patients.

	Unplanned Transfer (*n* = 31)	Control (*n* = 406)	*p*-Value
Age, yr	67.6 ± 11.5	65.4 ± 15.5	0.310
Sex (M:F)	22:9	212:194	0.624
Stroke type (infarct:hemorrhage)	19:12	258:148	0.848
Brain lesion (Supratentorial:Intratentorial:both)	22:7:2 *	172:188:46	0.008
Lesion side (right:left:bilateral)	13:10:8	186:149:71	0.508
Comorbidity			
Hypertension (*n*, %)	21 (67.7%)	232 (57.1%)	0.265
Diabetes mellitus (*n*, %)	14 (45.2%)	133 (32.8%)	0.171
Atrial fibrillation (*n*, %)	8 (25.8%) *	46 (11.3%)	0.040
Hyperlipidemia (*n*, %)	7 (22.6%)	127 (31.3%)	0.419
Functional status at admission to the department of physical and rehabilitation medicine			
K-MBI	17.3 ± 23.3 *	29.2 ± 26.4	0.015
MI-S	51.9 ± 29.7	59.7 ± 25.7	0.175
MI-U	53.0 ± 25.7	60.1 ± 27.4	0.222
MI-L	50.6 ± 30.2	59.2 ± 26.4	0.134
FAC	0.4 ± 1.1 *	1.1 ± 1.3	0.001
K-MMSE	10.3 ± 9.6 *	17.0 ± 10.7	<0.005

K-MBI, Korean version of the modified Barthel index; MI-S, side score of Motricity Index; MI-A, arm score of Motricity Index; MI-L, leg score of Motricity Index; FAC, Functional Ambulatory Category; K-MMSE, Korean Mini-Mental Status Examination. * *p* < 0.05, compared with the control group.

## Data Availability

The data that support the findings of this study are available from the corresponding author upon reasonable request.

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
