# Peer review of "Factors Associated with Unplanned Transfer of Patients with Brain Tumor from Inpatient Rehabilitation Unit to Primary Acute Care Units"

_jpm, 2023, doi:10.3390/jpm13010131_

Round 1
Reviewer 1 Report
well organized paper with interesting results.
Author Response
Response to Reviewer 1 Report
well organized paper with interesting results.
Response to general comments: First, we appreciate the reviewer’s comments.
Reviewer 2 Report
The paper Gyoung Ho Nam et al entitled “Factors Associated with Unplanned Transfer of Patients with Brain Tumor from Inpatient Rehabilitation Unit to Primary Acute Care Units ” aims to was to investigate the incidence of patients with brain tumor with an unplanned transfer to the primary acute care department during a comprehensive inpatient rehabilitation and factors associated with such unplanned transfer. I have several comments as following.
1. It is not clear why the authors choose stroke patients as control.
2. the type of brain tumor would be an important contributor to the transfer, the authors should consider adding this potential factor.
3. The significance of the study is explained not very well. The authors may need to emphasize that a bit.
4. for the statistical analysis, Student’s t-test was used for numerical variables of normal distribution, and the non-normally distributed numerical variables were analyzed with non-parametric test (Wilcoxon rank-sum test). The authors should make it clear in this study.

Author Response
Response to Reviewer 2 Report
The paper Gyoung Ho Nam et al entitled “Factors Associated with Unplanned Transfer of Patients with Brain Tumor from Inpatient Rehabilitation Unit to Primary Acute Care Units” aims to was to investigate the incidence of patients with brain tumor with an unplanned transfer to the primary acute care department during a comprehensive inpatient rehabilitation and factors associated with such unplanned transfer. I have several comments as following.
Response to general comments: First, we appreciate the reviewer’s insightful comments. The manuscript has been revised with adequate clarity to address the specific comments. More specifically, the manuscript has been revised to comply with the reviewer’s detailed comments, as follows.
- It is not clear why the authors choose stroke patients as control.
Response: Among patients with various brain disorders such as stroke, brain tumor, Parkinson's disease, and traumatic brain injury, the effects of comprehensive inpatient rehabilitation on patients with subacute stroke have the clearest evidence. In addition, our previous study [1] reported that the effects of comprehensive inpatient rehabilitation in patients with subacute stroke and brain tumor were similar. Therefore, in this study, patients with subacute stroke were chosen and analyzed as a control group for patients with brain tumor.
We added above descriptions in the discussion section of the amended manuscript.
Ref.> 1. Lee, H.S.; Yeo, S.; Kim, Y.-H.; Chang, W.H. Short-Term Effects of Intensive Inpatient Rehabilitation in Patients with Brain Tumor: a Single-Center Experience. Brain Neurorehabil 2018, 11.
- the type of brain tumor would be an important contributor to the transfer, the authors should consider adding this potential factor.
Response: We totally agreed with the review’s comments. Consistent with your comments, the incidence of unplanned transfer in patients with brain tumor was significantly higher in the primary tumor. This result was more clearly described in the results and discussion sections of the amended manuscript. Unfortunately, the further analysis of the causes was difficult to perform in this analysis due to the relatively small number of study participants. Thank you again for your good points.
- The significance of the study is explained not very well. The authors may need to emphasize that a bit.
Response: Thank you for your kind comments.
These results could be useful to identify patients with brain tumor at higher risk for unplanned transfer to the primary acute care units and to guide physiatrists for the closer supervision during the inpatient rehabilitation in patients with brain tumor.
We added above descriptions in the discussion section of the amended manuscript.
- for the statistical analysis, Student’s t-test was used for numerical variables of normal distribution, and the non-normally distributed numerical variables were analyzed with non-parametric test (Wilcoxon rank-sum test). The authors should make it clear in this study.
Response: We described more detail in the amended manuscript as “The Shapiro-Wilk test was used to determine the distributional normality of all continu-ous variables (all were found to be normally distributed; p>0.05).”
Reviewer 3 Report
The manuscript by Nam et al describes the higher incidence of transfer to primary acute care units of patients with brain tumor compared to stroke patients during intensive inpatient rehabilitation.
Before publication the manuscript should be improved as follow.
1. A recent paper, taking in consideration the same clinical aspect, should be mentioned (Bhakta et al 2022).
2. What does the following sentence mean? “…brain tumor itself causes problems such as tumor progression and metastasis…” The brain tumors as glioma are known to be not metastatic.
3. In figure 1 the standard deviation is missed. The authors should show it in a small table.
4.The authors should better describe the data in figure 2 after 20 days per each group.
5. It is important to take in consideration the tumour grade that should be an important evaluation factor for the clinical outcome.
Author Response
Response to Reviewer 3 Report
The manuscript by Nam et al describes the higher incidence of transfer to primary acute care units of patients with brain tumor compared to stroke patients during intensive inpatient rehabilitation.
Before publication the manuscript should be improved as follow.
Response to general comments: First, we appreciate the reviewer’s insightful comments. The manuscript has been revised with adequate clarity to address the specific comments. More specifically, the manuscript has been revised to comply with the reviewer’s detailed comments, as follows.
- A recent paper, taking in consideration the same clinical aspect, should be mentioned (Bhakta et al 2022).
Response: Thank you for your appropriate comments. The article by Bhakta et al. 2022 [1] is considered to be s similar aspect to our manuscript. We added this article as one of reference in the amended manuscript as “Bhakta el al. recently reported the incidence of unplanned transfer in patients with brain tumor as 34%. They reported the incidence and factors of unplanned transfer ac-cording to the type of cancer. However, the comparison with subacute stroke, which is the target disorder of intensive inpatient neurorehabilitation, has not been reported to the best of our knowledge
Ref.> 1. Bhakta, A.; Roy, I.; Huang, K.; Spangenberg, J.; Jayabalan, P. Factors associated with unplanned transfers among cancer patients at a freestanding acute rehabilitation facility. PM R 2022, 14, 1037-1043, doi:10.1002/pmrj.12681.
- What does the following sentence mean? “…brain tumor itself causes problems such as tumor progression and metastasis…” The brain tumors as glioma are known to be not metastatic.
Response: Thank you for your careful comments. We changed this sentence to “Because brain tumor itself causes problems such as tumor progression or metastasis,”
- In figure 1 the standard deviation is missed. The authors should show it in a small table.
Response: Incidence of unplanned transfers in Fig. 1 was a variable without SD, because it calculated the proportion of unplanned transfer that occurred in each group. There seems to have been some confusion about the term rate in Fig 1. Terminology has been sorted out. Sorry for the confusion.
- The authors should better describe the data in figure 2 after 20 days per each group.
Response: Thank you for your insightful comments. Most of participants in this study were discharged or had a planned transfer within 3 weeks. Therefore Fig. 2 could only be drawn within 30 days. Table 1 was supplemented by adding the duration of inpatient rehabilitation.
- It is important to take in consideration the tumour grade that should be an important evaluation factor for the clinical outcome.
Response: We totally agreed with your comments. If the analysis is performed for patients with primary brain tumor, we think that the part pointed out by reviewer should be considered as a very important factor. However, the tumor grade was not evaluated because this study included various brain tumors such as metastatic or hematologic brain tumors in addition to primary brain tumors. The further study with only patients with primary brain tumor patients will be supplemented through additional research. This part was added to limitation as “In this study, patients with brain tumor were classified only as those with primary, hematologic, and metastatic tumors. Therefore, we could not investigate each pathologic type or grade of tumor in detail.
Round 2
Reviewer 2 Report
The authors sufficiently addressed my concerns.